# Susceptibility of Oilseed Radish (*Raphanus sativus* subsp. *oleiferus)* Cultivars and Various Brassica Crops to *Plasmodiophora brassicae*

**DOI:** 10.3390/pathogens13090739

**Published:** 2024-08-29

**Authors:** Ann-Charlotte Wallenhammar, Eva Edin, Anders Jonsson

**Affiliations:** 1Rural Economy and Agricultural Society, HS Konsult AB, Gamla vägen 5G, SE-702 22 Örebro, Sweden; 2Rural Economy and Agricultural Society, HS Konsult AB, Brunnby Gård 1, SE-725 97 Västerås, Sweden; eva.edin@hushallningssallskapet.se; 3RISE Research Institutes of Sweden AB, Box 187, SE-532 32 Skara, Sweden

**Keywords:** clubroot, oilseed rape (*Brassica napus*), radish (*Raphanus sativus* var. *sativus*), early garden turnip (*Brassica rapa* var. *rapa)*, white mustard (*Sinapis alba*), May turnip (*Brassica rapa* subsp. *rapa* f. *majalis*)

## Abstract

Oilseed radish (OR; *Raphanus sativus* var. *oleiferus*) is grown as a cover crop and develops a unique taproot, absorbing nitrogen left by the previous crop. The aim of this project was to investigate the resistance of OR cultivars (cvs.) to *Plasmodiophora brassicae*, the causal agent of clubroot disease. Twelve market cvs. were compared with cvs. of clubroot-resistant (CR) winter oilseed rape (OSR; *Brassica napus*) and other selected species of the Brassicaceae family. The study was performed as a replicated bioassay in a growth chamber using a specially composed mixture of field soils holding the natural inoculum of *P. brassicae*. The results show that the OR cultivars were infected, which implies that OR multiplies the pathogen. The susceptibility of the OR cultivars was not significantly different from that of the CR OSR cultivars Alister and Archimedes, but it was significantly different from that of the OSR cv. Mendel. The disease severity index (DSI) for OR cultivars ranged from 2.3 to 9.3, and disease incidence was 3–17%. The best performance was shown by black radish (*Raphanus sativus* var. *niger*) with a DSI of 0.3. For sustainable brassica crop production, we suggest avoiding OR as a cover crop in crop rotations, including OSR or other brassica crops, since there is a risk of increasing inoculum in the soil.

## 1. Introduction

Cover crops (CCs; species mixtures that are grown between two cash crops) can deliver multiple benefits for farmers in their crop production by bringing resilience into their agricultural system and providing several ecosystem services. Cover crops are grown with the purpose of prolonging the period of growth, preventing the leaching of nutrients, improving soil structure, limiting soil erosion, providing synergies to enhance biodiversity and carbon sequestration [1,2,3,4,5], and managing weeds due to the soil coverage of the leaves and allelopathic ability [6,7]. Brassicaceae or crucifer CCs have the potential to suppress pathogens and weeds due to the release of several potentially biocidal hydrolysis products, such as isothiocyanates [8,9]. Oilseed radish (OR; *Raphanus sativus* subsp. *oleiferus*) is often grown as a CC, in pure stands or in mixtures with other plant species, reducing soil erosion, improving retention of nutrients and soil structure, well as reducing plant parasitic nematode populations [1,3,4,7,10]. Nematode resistant OR cultivars are able to decrease the population of sugar beet cyst nematodes (*Heterodera schachtii*), potato cyst nematodes (*Globodera* spp.), and root-gall nematodes (*Meloidogyne* spp.) in soil by biofumigation or serve as a poor host that disables the formation of cysts/galls [10,11,12,13,14].

Clubroot, caused by the soilborne obligate pathogen *Plasmodiophora brassicae*, is one of the most devastating soil-borne diseases of brassica oilseed and vegetable crops, capable of causing significant crop losses. The control of clubroot is particularly difficult due to the long persistence of the resting spores [15], which impedes disease control by means of crop rotation. The intensified cultivation of oilseed rape (OSR, *Brassica napus* L.) and of cruciferous vegetable crops has led to an increasing proportion of clubroot-infected arable land in many European countries [16,17,18] and worldwide, e.g., Canada, Australia, USA, China, India, South Africa, Mexico, and other countries in Latin America [19,20,21,22,23,24,25,26]. Consequently, scientific studies have increased to great numbers during the last decade [27,28].

The typical clubroot symptoms consist of an enlarged root that contains tens of millions of resting spores per gram of root, and only 100–1000 spores per g of soil are needed for an infection [29]. The life cycle is complex and is defined by three stages. The resting spores surviving in the soil (i) are robust with a well-developed mechanism of different spore walls that provide protection against other microorganisms. At high soil moisture, after 18 h of exposure, cabbage seedlings are infected [30], and at optimal temperatures ranging from 18 °C to 24 °C, resting spores germinate. (ii) Primary zoospores are produced, infecting the root hairs, and an intracellular primary phase takes place and is followed by the release of (iii) secondary zoospores [31]. These spores infect the rhizodermal cells of the root, and the pathogen subsequently induces local hypertrophy caused by mis-regulated and plant-derived hormonal pathways [32]. Due to their long persistence, the spores remain in the soil for up to almost 20 years [15], and the control of clubroot is particularly difficult. The most successful management strategy of clubroot, once it is established in a field, is the cultivation of clubroot-resistant (CR) cultivars of OSR and other brassica crops. It is worth mentioning that CR cultivars, in fact, are partially resistant, as disease symptoms develop even at low spore density in the soil, and thereby, the pathogen inoculum is multiplied in infested fields [20,21].

Radish (*Raphanus sativus*) belongs to the Brassicaceae family, originating in China and along the coastal regions from the Mediterranean region to the Black Sea [33]. The most remarkable characteristic of radish is its tuberous roots, showing huge variations in shape and size [34]. Radish is an economically important root crop cultivated worldwide. Most radish cultivars and lines are highly resistant to *P. brassicae* [32], but some are susceptible, as field studies in the Netherlands and greenhouse studies in the US showed disease in *R. sativus* var *niger* and *R. sativus* subsp. *oleiferus* [14,35]. Oilseed radish is one of the most widely used cover crops and is considered generally resistant to clubroot (*P. brassicae*), but in trials, a variation was reported (https://www.joordens.com/en/cover-crops/oil-seed-radish/, accessed on 8 July 2024). Seeding brassica crops as CC or sanitation crops for biocontrol [36] might cause potential disservices, such as hosting diseases.

Oilseed radish has, in general, a high resistance, as previously shown in a study performed on selected cultivars of *R. sativus* subsp. *oleiferus* and of one cultivar of *R. sativus* var *longipennatus* [37]. Other CCs in the Brassicaceae family, such as white mustard (*Sinapis alba*) and brown mustard (*Brassica juncea*), are highly susceptible [35,38,39]. Turnips are popular root vegetables susceptible to *P. brassicae* and hence are included in this study. Given the high interest in growing CCs, it is important to increase our knowledge of the susceptibility to *P. brassicae* for the OR cultivars available and for other species in the Brassicaceae family.

In this study, the prerequisites for integrating OR as a CC in a crop rotation were evaluated using a mixture of Swedish field soils with a natural inoculum of *P. brassicae*. The aim was to develop a recommendation for OSR growers about including OR as a CC in a crop rotation with OSR as a cash crop.

Our general hypothesis was that OR develops disease symptoms to the same extent as clubroot-resistant winter OSR cultivars.

The hypotheses were as follows: (i) Oilseed radish has a high degree of resistance to *P. brassicae*, but there are variations among cultivars. (ii) Oilseed radish develops disease symptoms to the same extent as clubroot-resistant winter OSR cultivars. Moreover, (iii) oilseed radish as a cover crop has the potential to increase the prevalence of *P. brassicae* if incorporated into a crop rotation of oilseed rape.

## 2. Materials and Methods

### 2.1. Soil Preparation

Soil was collected from the top 20 cm of topsoil using a spade from fields with known high levels of infestation of *P. brassicae* in Tomelilla municipality in southern Sweden (T; 14% clay, pH 6.5), Örebro municipality in central Sweden (Ö; 15% clay, pH 5.9), and from a field where *P. brassicae* DNA was below the detection level in Skara municipality in central Sweden (S; 13% clay, pH 6.2). In a pilot project, a soil mixture was developed at an inoculum level, causing severe symptoms in susceptible OSR cultivars and clear symptoms in CR cvs. The Tomelilla soil was gathered from a field experimental site with a field abundance of 1,187,100 gene copies of *P. brassicae* per g of soil (corresponding to 2.7 million spores per g of soil) [40]. The pathotype is designated as 16/23/29 according to the European Clubroot Differential (ECD) and as P1 according to Somé et al. [17,41]. The soils were dried at room temperature (about 5% water content) and mixed in the following proportions: 18% (T), 10% (Ö), and 72% (S). After weighing, the soil mixture was placed in a concrete mixer (MEEC 63 L, 220 W, 27.5 rpm) and run for 10 min before the soil was distributed in pots.

### 2.2. Plant Material

The following species and cultivars were examined: four winter oilseed rape cultivars (*Brassica napus* L): Atora, a cultivar susceptible to clubroot, and three clubroot (CR)-resistant cultivars: SY Alister, LI Archimedes, and NPZ Mendel; twelve cultivars of oilseed radish (*Raphanus sativus* subsp. *oleiferus*): Angus, Atlantis, Cassius, Comet, Defender, Double max, Edwin, Farmer, Maximus, Merkur, Silentia, and Siletta Nova; two cultivars of radish (*Raphanus sativus* var. *sativus*): Rudi and Patricia; one cultivar of black radish (*Raphanus sativus* var. *niger*): Runder Schwartzer Winter; two cultivars of May turnip (*Brassica rapa* subsp. *rapa* var. *majalis*): Jaune boule d’Or (JB) and Petrowski; one cultivar of early garden turnip (*Brassica rapa* subsp. *rapa*); and one cultivar of white mustard (*Sinapis alba*): Mustang (Table 1).

### 2.3. Bioassay

The bioassays were carried out in a growth chamber with a completely randomized design and maintained for a six-week period to ensure optimal infection, according to Wallenhammar et al. [15]. The composite soil mixture was divided into in four pots (Göttinger, 9 × 9 × 9.5) with a volume of 0.5 L for each cultivar. The pots were watered to field capacity, and 15 seeds per cultivar were sown in each pot. After sowing, the pots were covered with plastic film, placed individually on pot plates with a diameter of 14 cm, closed with rubber bands, and incubated in a growth chamber with 14 h LED light (PAR Phillips LED) at 18–20 °C. After germination and when the seedlings reached the cotyledon stage, the plastic cover was removed, and they were thinned out to ten seedlings at true leaf stage. Watering and fertilization were performed according to a previously prepared protocol [43]. The bioassay was repeated three times from March to November 2019.

Infected plants were scored according to the following classes: 0 = no galls; 1 = enlarged lateral roots; 2 = enlarged taproot; 3 = enlarged napiform taproot; 4 = enlarged napiform taproot, lateral roots healthy; and 5 = enlarged napiform taproot, lateral roots infected (Figure 1). Disease severity index (*DSI*) was calculated as follows:$$DSI=\frac{\Sigma \;(\left(Class\;no\right)\times \left(No.\;of\;plants\;in\;each\;class\right))\times 100}{N\times (No.\;of\;classes-1)}$$
where *N* = the total number of plants. The proportion of roots with infection was calculated (disease incidence, DI).

### 2.4. Statistical Analyses

The results were processed using ANOVA tests followed by Tukey’s HSD-test (*p* < 0.05) in JMP 15.2 (SAS Inst. Inc, Cary, NC, USA). Interactions between the logarithm of the number of gene copies g^−1^ soil and yield, disease severity index, and disease incidence for the cultivars in the field trials and bioassays were calculated using regression analysis in JMP 15.2. All data were used in the regression analysis for the bioassays.

## 3. Results

The results of the bioassays are displayed in Table 2. The assessments showed that all OR cultivars were infected, and disease severity ranged from a DSI of 2.3 to 9.2 with no significant difference between OR cultivars. Clubroot-resistant cvs. Alister and Archimedes showed a similar infection (a DSI of 21.6 and 26.0, respectively), while the DSI was higher for cv. Mentor (45) and closer to susceptible cv. Atora (69.3), which is significantly different from the CR cultivars. White mustard (DSI of 76.4) and the *B. rapa* species were highly susceptible (DSI 66.6–75.4), while *R. sativus* var *sativus* cultivars showed intermediate infection (DSI of 18 and 12, respectively). *Raphanus sativus* var. *niger* was highly resistant with a DSI of 0.3.

## 4. Discussion

European agriculture is facing high expectations and pressure from society and policymakers to support environmental protection and climate change mitigation [2]. In a recent study of farm surveys, the authors conclude that policy is by far the strongest determinant of adoption rates and adoption intensities and that agronomic motives are a much weaker motive for adoption [44]. In light of these findings, it is of the utmost importance to emphasize the enhanced environmental and climate performance potential that farmers worldwide can contribute to and that choosing CCs needs a precise plan, adapted to the cash crops to be grown in a specific field.

In this study, we have focused on the potential species of the Brassicaceae family belonging to the genera *Brassicae*, *Raphanus*, and *Sinapis* to be used as CCs grown as sole crops or in mixtures with other plant families. Species of these genera are hosts of *P. brassicae* causing clubroot, and some are severely infected [37,44]. This study is, to our knowledge, the first study where CR-resistant and susceptible cultivars of winter OSR, OR, and white mustard are evaluated for clubroot susceptibility in a soil mixture from soils with natural inoculum of *P. brassicae.* We also included species from the *Raphanus sativus* var *sativus* and *Brassica rapa*, vegetable crops with increasing popularity, with some of the cultivars representing heirloom cultivars. Two OR cvs. in this study have been available for several decades, i.e., cvs. Silentia and Siletta Nova [14], while new OSR cultivars are continuously introduced.

The clubroot disease is a serious threat to OSR production worldwide, and the increasing proportion of clubroot-infected arable land in Sweden has serious consequences for OSR growers [45]. Attractive market prices have contributed to expanding production in recent years, often through increased OSR frequency within rotations. The most successful management strategy of clubroot, once it is established in a field, is the cultivation of CR cultivars of OSR and other CR brassica species [39,46].

We examined a total of 23 different cultivars of different species within the plant family Brassicaceae in a specially composed soil mixture holding a natural inoculum of *P. brassicae* that caused high disease severity indices (DSIs). Our results are in accordance with hypothesis (i), that oilseed radish has a high degree of resistance to *P. brassicae*, but there are variations among cultivars. Hypothesis (ii) that OR and CR winter OSR cultivars were infected to the same extent is also in accordance with the results, since all OR cultivars showed clubroot symptoms with a variation in DSI between 2.5 and 9.2 on average, lower than in CR winter OSR cultivars. However, there were no statistically significant differences between CR OSR cultivars and OR cultivars. Winter OSR cv. Mendel was severely infected with an average DSI of 45, which is above the cut-off point of a DSI of 25 used to classify plant reactions as resistant or susceptible [17]. The disease severity of cv. Mendel is higher than determined in a previous bioassay [39] using soil from the same field, but at high inoculum densities, resistance may be overcome by a less virulent pathogen [46].

To conclude, our results are in accordance with hypothesis (iii) that OR as a CC has the potential to increase the prevalence of *P. brassicae* if incorporated in a crop rotation where the pathogen is prevalent. Oilseed radish is an autumn CC seeded in early autumn to grow in the fallow period between the two main cash crops. The crop is not harvested, and the biomass remains on or in the soil [36,47]. Thus, the time point of seeding coincides with the time point of seeding winter OSR, when conducive conditions for infection of *P. brassicae* are likely to occur concerning soil temperature and soil moisture.

Integrating strategies and crop management will reduce but not eliminate this disease [39,48]. Taking into account the fact that *P. brassicae* can infect other plant species outside the Brassicaceae family serving as hosts of the pathogen, e.g., *P. brassicae* DNA was recently detected in roots of field pea (*Pisum sativum)* and in roots of phacelia (*Phacelia tanacetifolia*) [49], whereas the germination and death of *P. brassicae* resting spores were promoted by root exudates of marigold (*Tagetes ercta*) [50].

The tested cultivars of different turnips and white mustard were susceptible. The black radish cultivar Runder Schwarzer Winter, a German heirloom cultivar from the 1850s (DSI of 0.3), showed the lowest infection. The disease severity indices of the radish cultivars were intermediate, similar to the DSI of CR OSR cv. Alister, and were considered as resistant [17]. Radishes are a vegetable crop with a long history, which evolved a large and diverse range of germplasm resources [51]. A recent study in China [52] identified 56 clubroot-disease-resistant resources, most of them derived from China, emphasizing the richness of disease-resistant resources in radishes compared to other cruciferous crops such as OSR and cabbage that lack clubroot-resistant germplasm.

The commercial cultivars of OR tested here are considered to provide high resistance, but since they develop clubroot symptoms, we generally suggest to avoid growing OR in crop rotations with brassica crops due to an increased risk of propagating *P. brassicae*. Highly resistant lines of OR were found in German assays, which are suitable for breeding highly resistant cultivars used as CCs [53]. The resistance to *P. brassicae* is highly dependent on the virulence of the pathotype [46]. Many CR cultivars, including Mendel, developed symptoms when inoculated with pathotypes from central Europe and Sweden [17].

In a previous study, the agronomic performance of CR cultivars of winter OSR [39] was studied, and it was clear that at high inoculum densities, disease severity was higher, and in bioassays, the resistance was eroded. In field-specific conditions, OR is seeded after the cash crop is harvested, during a period of high soil temperatures in August, and short and heavy rainfall will provide optimal conditions for infection.

Sweden has a very long tradition of growing plants that are susceptible to clubroot. Different kinds of swedes and turnips were very important elements in the north European diet before potatoes became an important component of the staple food [45], where the turnip “svedjerova” (*B. napus* var *napus* Thell), an old Finnish cultivar of turnips, dates back to the 13th century (https://merakiseeds.com/turnip-svedjerova accessed on 8 July 2024). This cultivar was traditionally grown by Finnish and Swedish people and was, hence, according to our study, highly susceptible to clubroot. Later, different cultivars of May turnips were grown, often on arable land. Turnips were susceptible in this study, and we know that clubroot was a widely known problem in turnips and swedes prior to the onset of the Swedish OSR production in the 1940s [45]. Over time, the pathogen has spread and multiplied slowly in the fields, and after 10–12 OSR crops in the rotation, clubroot caused large yield losses in long-term field fertility trials in southern Sweden [54]. From our own and international studies, it is known that the pathogen can be spread by soil on machines, boots, and seeds, as well as through soil erosion, wind, and water runoff [1,55].

Cover crops have been a part of the European common agricultural policy (CAP) 2014–2020 [43]—which introduced subsidies for European farmers growing CCs. In the current CAP 2023–2027, subsidies for growing CC with the purpose of carbon sequestration are added. Within the regulations for CC subsidies, several suggested CC species belong to the Brassicaceae family that can be infected by *P. brassicae*, such as oilseed radish, spring oilseed rape, spring turnip rape, radish, and white mustard, where white mustard has already caused severe problems [Adholm, per comm], and in this study, the cultivar tested was highly susceptible.

Further studies are needed to evaluate the OR in different pathotype populations of *P. brassicae*, as, in a recent study, 42 different pathotypes were designated from isolates of winter OSR in central Europe and Sweden [17]. This study also provided evidence for a shift towards increased virulence in *P. brassicae* populations compared to previous studies [46]. Furthermore, in a German pathogenicity test where a range of plant species were tested, *R. sativus* was not infected by a *P. brassicae* isolate virulent on a susceptible OSR cultivar nor by an isolate virulent also on cv. Mendel [49], which is divergent from the results presented in this study. Local conditions will play an important role in avoiding the multiplication of the pathogen inoculum while selecting the proper species and cultivars of both the main crop and CC, with respect to the crops grown in the rotation.

The germination of resting spores is stimulated by the soil microbial community, and the evidence of a germination stimulant by host plants was found by Macfarlane, 1970 [56], for *P. brassicae* resting spores. Recent studies reveal that several enriched bacterial taxa in the stimulating community are significantly correlated with spore germination rates and are suggested to be involved as stimulation factors [57]. The importance of the diversity of rhizosphere microbial community, revealed by high-throughput sequencing, showed that OSR planted after soybeans (*Glycine maximus*) significantly increased the population density of microbes that could inhibit *P. brassicae*. Conversely, a cruciferous crop grown after another cruciferous crop significantly increased plant pathogens, including *P. brassicae*, *Olpidium* sp., and *Colletotrichum* sp. [58]. Thus, the success rate of germination of resting spores is dependent on the crops grown in a rotation. Improved soil health related to the composition of soil microbial communities was also reported by [50] for the marigold–Chinese cabbage rotation compared to Chinese cabbage in monoculture.

The great interest in increasing carbon sequestration, as well as reducing plant parasitic nematodes in arable land by growing Brassica CCs, generally requires knowledge of the potential risk of multiplying soil-borne pathogens. Furthermore, an increased knowledge of the importance of the soil ecological environment for plant–microorganism interactions in the plant rhizosphere needs to be further explored and implemented.

## 5. Conclusions

Cover crops can deliver multiple benefits for farmers in their crop production, but potential disservices, such as disease hosting, must be taken into consideration. Oilseed radish, often used as a CC, developed symptoms of clubroot disease in the bioassays. To avoid potential propagation of *P. brassicae*, OR or other species within the Brassicaceae family should not be included in the crop sequence when the cash crop is OSR or other species within the Brassicaceae family. Managing clubroot once it is established in a field is very difficult due to the persistence of the pathogen in soil. Proactive measures, such as soil testing based on DNA technology, must be used to a greater extent to avoid outbreaks in fields grown with susceptible cultivars and to prevent consistent loss of valuable seed and crop yield. It is evident that special care must be taken in the CC selection of species in mixtures or as sole crops to avoid pathogen increase.

## Figures and Tables

**Figure 1 pathogens-13-00739-f001:**
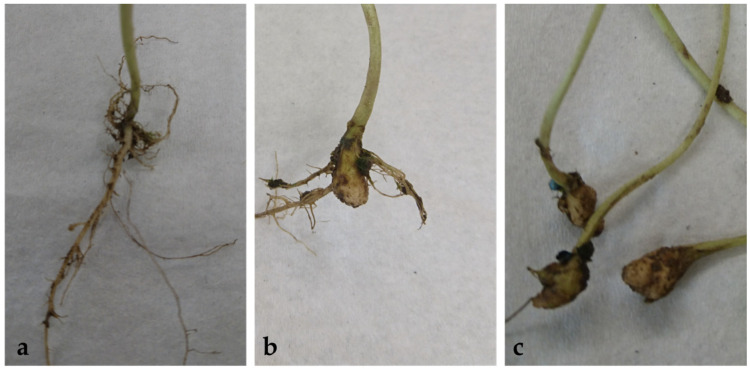
Characteristics of scoring of assessment of clubroot on plant roots in bioassays are as follows: (**a**) enlarged lateral roots (class 1); (**b**) enlarged napiform taproot, lateral roots infected (class 5); (**c**) enlarged napiform taproot, lateral roots infected (class 5).

**Table 1 pathogens-13-00739-t001:** Species, cultivars, and characteristics of plants used in three bioassays performed in a growth chamber using a mix of soil from naturally infested soils and non-infested soils with a susceptible (S) winter oilseed rape (OSR) cultivar and three clubroot-resistant OSR cultivars, oilseed radish, radish, black radish, May turnip, and early garden turnip.

Botanical Name	Common Name	EPPO Code	Plant Type	Cultivar	Nematode Resistance	Source of Nematode Resistance	Seed Provider
*Brassica napus*	Winter oil seed rape	BRSNW	Cash crop	Alister (CR)			Syngenta, Malmö, Sweden
				Archimedes (CR)			Gullviks, Malmö, Sweden
				Mendel (CR)			Lantmännen Lantbruk, Malmö, Sweden
				Atora (S.)			Lantmännen Lantbruk, Malmö, Sweden
*Raphanus sativus* subsp. *oleiferus*	Oilseed radish	RAPSO	Cover crop	Angus			PH Pedersen, Lundsgaard, Germany
				Atlantis			PH Pedersen, Lundsgaard, Germany
				Cassius			PH Pedersen, Lundsgaard, Germany
				Comet	*Heteroderea schachtii*	P.H.Petersen.com	PH Pedersen, Lundsgaard, Germany
				Defender	*Meloidogyne chitwoodi*	[13]	PH Pedersen, Lundsgaard, Germany
				Double max			PH Pedersen, Lundsgaard, Germany
				Edwin			Agortus AB, Svalöv Sweden
				Farmer			PH Pedersen, Lundsgaard, Germany
				Maximus	*M. chitwoodi*	[13]	PH Pedersen, Lundsgaard, Germany
				Merkur			PH Pedersen, Lundsgaard, Germany
				Silentia	Trap crop for *M. hapla*	[42]	PH Pedersen, Lundsgaard, Germany
				Siletta Nova			PH Pedersen, Lundsgaard, Germany
*Raphanus sativus* var. *sativus*	Radish	RAPSR	Vegetable crop	Patricia			Lindbloms Frö, Hammenhög Sweden
				Rudi			Lindbloms Frö, Hammenhög Sweden
*Raphanus sativus* var. *niger*	Black radish	RAPSN	Vegetable crop	Runder Schwartzer Winter			Lindbloms Frö, Hammenhög Sweden
*Brassica rapa* subsp. *rapa* var. *majalis*	May turnip	BRSRR	Vegetable crop	Jaune Boule d´Or			Lindbloms Frö, Hammenhög Sweden
				Petrowski			Lindbloms Frö, Hammenhög Sweden
*Brassica rapa* subsp. *rapa*	Early garden turnip	BRSRR	Vegetable crop	Svedjerova			Lindbloms Frö, Hammenhög Sweden
*Sinapis alba*	White mustard	SINAL	Cover and cash crop	Mustang			Olssons Frö, Helsingborg, Sweden

**Table 2 pathogens-13-00739-t002:** Disease severity index (DSI) and disease incidence (DI, percentage of diseased plant) of clubroot on average for three bioassays performed in a growth chamber using a mix of soil from naturally infested soils and non-infested soils with a susceptible (S) winter oilseed rape (OSR) cultivar and three clubroot-resistant OSR cultivars, oilseed radish, radish, black radish, May turnip, and early garden turnip (*n* = 4).

Botanical Name	Cultivar	Disease Severity Index (0–100)		Disease Incidence (0–100)	
*Brassica napus*	Alister (CR)	21.6	cdb	34.1	cde
	Archimedes (CR)	26.0	bc	38.1	cd
	Mendel (CR)	45.0	b	56.6	bc
	Atora (S)	69.3	a	76.9	ab
*Raphanus sativus* subsp. *oleiferus*	Angus	4.2	cd	11.1	def
	Atlantis	8.2	cd	13.3	def
	Cassius	5.5	cd	12.1	def
	Comet	5.7	cd	14.9	def
	Defender	9.2	cd	17.7	def
	Double max	2.5	cd	4.8	de
	Edwin	4.0	cd	9.8	def
	Farmer	2.9	cd	4.3	ef
	Maximus	2.3	cd	3.2	f
	Merkur	2.6	cd	6.6	def
	Silentia	5.1	cd	10.2	def
	Siletta Nova	2.5	cd	4.2	ef
*Raphanus sativus* var. *sativus*	Patricia	17.7	cd	31.6	cde
	Rudi	12.4	cd	20.8	def
*Raphanus sativus* var. *niger*	Runder Schwartzer Winter	0.3	d	0.8	f
*Brassica rapa* subsp. *rapa* var. *majalis*	Jaune Boule d’Or	66.4	a	72.4	ab
	Petrowski	75.4	a	81.8	ab
*Brassica rapa* subsp. *rapa*	Svedjerova	67.6	a	73.1	ab
*Sinapis alba*	Mustang	76.4	a	88.9	a
*p*-value		<0.001		<0.001	
Coefficient of variance		60.5		61.6	

Different letters indicate significant differences according to Tukey’s HSD-test (*p* < 0.05).

## Data Availability

Data presented in this study are available by the authors upon request.

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
