# Peer review of "Susceptibility of Oilseed Radish (Raphanus sativus subsp. oleiferus) Cultivars and Various Brassica Crops to Plasmodiophora brassicae"

_pathogens, 2024, doi:10.3390/pathogens13090739_

Round 1

Reviewer 1 Report

Comments and Suggestions for Authors

The paper presented by Ann-Charlotte et al. described a survey of various oilseed radish, radish, turnip and oilseed rape materials when confronted with soil-borne clubroot pathogens from Sweden. I think the clubroot disease is of great importance for crucifer plants worldwide, since a lot of crucifer crop species contribute substantial cash crop proportions in e.g. Australia, Canada and some European and Asian countries. However, although the data is quite clear, the overall manuscript is over-simplified. It sounds more like a fast-communication instead of a research paper. I would like to share some thoughts with the authors that might improve the general interest of the paper for scientific audience.

1.       Just one Table as the only data source is too simple. Table 1 described mainly the disease phenotypes for various materials with dif. Genetic background. Perhaps include some photo/pictures, especially for DSI 0-5, or at one particular stage that can show us how susceptible B. rapa can be compared to e.g. Black radish, that is highly resistant? Although numbers are good, phenotypic observation is always highly appreciated.

2.       Why not do separate field-collected pathogen infections in separate design? Since as I see the northern, middle and southern Swedish locations suffer different degree of clubroot due to e.g. Temperature and humidity and perhaps, pH value from the top 20cm soil collections. A mixture of the three soil collections would make it difficult to compare the effect of possible pathogen composition difference between the three soil samples, since some of the more susceptible host could have different DSI and IR (incidence rate) by infection of the T or S or Ö samples. I personally think this is also an interesting point, although without single spore isolates it is for now difficult to say how different they are, besides all being the same ECD and Some pathotype.

3.       Did the author collect fresh weight or any other agronomy traits with indication for the outcome of the clubroot disease, e.g. 20 or 30 days after infection? Could provide alternative data support for the main conclusion.

4.       Do we know any genomic information for the difference shown in Table 1, e.g. why Black radish is so resistant, on genome level or on CR loci presence, is there any explanation why they stand out as of particular resistant to clubroot? Could discuss at least.

5.       Is the 45 DSI value of Mendel consistent with previous findings in literature? Since Mendel is widely used in Europe as a moderate good clubroot resistant cultivars or?

Reviewer 2 Report

Comments and Suggestions for Authors

The manuscript describes an experiment on testing crops of Brassica,  Raphanus and Sinapis with a mix of soil from naturally infested soils with clubroot. The experiment design is straightforward. The results summarized in only one Table are clear.  Only minor revisions are needed if the journal editor thinks the scope of research and amount of efforts for study meet the journal requirements.

1.      Line 12: A “.” is required.

2.      Line 96:  Please clarify “1 187 1000 gene copies:

Reviewer 3 Report

Comments and Suggestions for Authors

Dear Authors,

In my opinion, the following points need your attention.

L15, L61, L174 (and possibly elsewhere): the concepts of the "genus Brassica" and "plant family Brassicaceae" seem to be mixed up. Please address.

L69: disservice = side-effect?

L100: concrete

I suggest showing typical photos for the disease gradient, explained in L134-140.

I suggest inclusion of data in the discussion on how non-Brassicaceae species and crop rotations change the soil abundance of P. brassicae or the disease incidence. Example papers include [1-3].

Best regards.

References

[1] https://doi.org/10.1016/J.EJA.2015.07.007

[2] https://doi.org/10.3390/plants11172295

[3] https://doi.org/10.1016/s2095-3119(20)63186-0
